# Plant-derived Pembrolizumab in conjugation with IL-15Rα-IL-15 complex shows effective anti-tumor activity

Kaewta Rattanapisit[1], Pipob Suwanchaikasem[1]*, Christine Joy I. Bulaon[1], Shiying Guo[2], Waranyoo Phoolcharoen[3,4]*

1 Baiya Phytopharm Co., Ltd., Bangkok, Thailand, 2 GemPharmatech Co., Ltd, Nanjing, China, 3 Center of Excellence in Plant-produced Pharmaceuticals, Chulalongkorn University, Bangkok, Thailand, 4 Department of Pharmacognosy and Pharmaceutical Botany, Faculty of Pharmaceutical Sciences, Chulalongkorn University, Bangkok, Thailand

⊚ These authors contributed equally to this work.
* Pipob.s@baiyaphytopharm.com (PS); Waranyoo.P@chula.ac.th (WP)

## Abstract

Anti-programmed cell death 1 (PD-1) monoclonal antibodies (mAbs) have proven to be effective in treating various cancers, including colorectal, lung, and melanoma. Despite their clinical success, some patients develop resistance to mAbs, requiring co-treatments with radio- or chemotherapy. Interleukin-15 (IL-15) is an immunostimulatory cytokine that promotes immune cell production and proliferation. It has been combined with mAbs and other immunotherapies to improve efficacy and reduce side effects. Fusion of anti-PD-1 mAb and IL-15 streamlines drug administration and management. In this study, we developed a prototype by conjugating the IL-15 receptor subunit alpha (IL-15Rα) and IL-15 complex to the C-terminus of anti-PD-1 Pembrolizumab (Pembrolizumab-IL-15Rα-IL15) using plant molecular farming for production. LC-MS revealed the presence of plant N-glycans (GnGnXF, GnXF and Man9GlcNAc2) on the molecule, which may affect receptor-binding avidity. However, ELISA demonstrated comparable binding efficacy of Pembrolizumab-IL-15Rα-IL15 to human PD-1 protein as commercial Pembrolizumab. In a mouse anti-cancer study, Pembrolizumab-IL-15Rα-IL15 (3 mg kg$^{-1}$) exhibited slightly improved tumor-growth inhibition, reducing tumor size by 94% compared to commercial Pembrolizumab (5 mg kg$^{-1}$) with an 83% reduction, regardless of statistically significant difference. In conclusion, Pembrolizumab-IL-15Rα-IL-15 was successfully produced using plant molecular farming and shows promise in addressing mAb drug resistance and enhancing the immunomodulatory effects of IL-15 payload.

## Introduction

Immunotherapy has emerged as a transformative approach in the treatment of cancer treatment [1]. Among commercially available immunotherapeutic agents, monoclonal antibodies (mAbs) and cytokines have shown considerable promise [2, 3]. They have been used either in

**Funding:** National Research Council of Thailand (NRCT) and Chulalongkorn University (Grant number: N42A670577) and the Thailand Science Research and Innovation Fund Chulalongkorn University (Grant number: BCG66330001) K.R. was supported by the National Science, Research, and Innovation Fund (NSRF) via the Program Management Unit for Human Resources & Institutional Development, Research, and Innovation (Grant number: B13F660137).

**Competing interests:** W.P. is a co-founder of Baiya Phytopharm Co., Ltd., Thailand. K.R., P.S., and C.J.I.B. are employees of Baiya Phytopharm Co., Ltd. S.G. is an employee of GemPharmatech Co., Ltd. The remaining author has no competing financial interests. There are no relevant non-financial interests to declare.

combination with conventional therapies, such as radiotherapy and chemotherapy, or as a second-line options when standard treatments fail [4, 5]. One example, Pembrolizumab (Keytruda), a humanized mAb, is approved for treating various cancer types, including melanoma, non-small cell lung cancer (NSCLC), head and neck squamous cell carcinoma (HNSCC) and triple-negative breast cancer (TNBC) [6]. It works by binding to programmed cell death protein 1 (PD-1) on T-cell lymphocytes, thereby inhibiting interaction between PD-1 on T cells and PD ligand 1 (PD-L1) on cancer cells. This immune checkpoint blockade allows immune cells to attack tumors [7].

Cytokines, such as interleukin-2 (IL-2), also play a crucial role in cancer immunotherapy. IL-2 has been approved for the treatment of melanoma and renal cell carcinoma [8]. It binds to IL-2 receptors (IL-2R) to activate growth and proliferation of various lymphocytes, including cytotoxic T cells, memory T cells, regulatory T cells ($T_{reg}$ cells), B cells and natural killer (NK) cells [9]. However, therapeutic IL-2 is known to cause unfavored side effects of vascular leak syndrome and activation-induced cell death, which limit its clinical utility [10]. IL-15, a cytokine with a similar role to IL-2, does not bind to IL-2R subunit α (IL-2Rα), mainly located on the surface of $T_{reg}$ cells [11]. In contrast, IL-15 binds specifically to IL-15R subunit α (IL-15Rα), located on the surface of antigen-presenting cells, such as dendritic cells and macrophages, to further inducing T and NK cell functions. By fusing IL-15Rα to IL-15, the complex can readily interact with IL-15Rβ and IL-15Rγ, located on the T and NK cells, without primarily binding to the antigen-presenting cells. Therefore, IL-15 in conjugation with IL-15Rα has been explored as a promising alternative and increasingly studied to become a new immunotherapeutic agent [12]. Despite these advances, current immunotherapies face several challenges that hinder their broader application and effectiveness. One major issue is the occurrence of refractory or relapsed disease, even after initial positive responses to checkpoint inhibitors drugs [13]. For instance, many patients exhibited limited efficacy due to inherent or acquired resistance to these therapies. Moreover, cytokines suffer from short half-lives and off-target binding, which further restrict their clinical use [12]. Attempts to improve cytokine performance through structural modifications and amino acid-sequence mutations have achieved varying success [14, 15].

There is a pressing need for innovative strategies to overcome the drawbacks of current immunotherapies. One promising method is the development of fusion molecules that combine mAbs with therapeutic cytokines into a single entity. This strategy simplifies the administration protocol, potentially enhances therapeutic efficacy and alleviates adverse events [5, 16, 17]. The use of fusion molecules is particularly compelling in combination therapies, which have shown synergistic effects in clinical trials [18]. For example, the combination of the IL-15-IL-15Rα-Fc complex (Anktiva) with Pembrolizumab is currently in phase II/III clinical trials for the treatment of stage IV and recurrent NSCLC (NCT05096663). However, this co-treatment still requires the separate administration of Pembrolizumab and IL-15-based molecules. To address this, fusion proteins have been increasingly developed, many of which have demonstrated superior anti-tumor activity in pre-clinical studies [19–22]. These molecules vary in their designs, with different linkage positions of IL-15 to Pembrolizumab, the use of diverse linkers, and specific amino acid mutations within the IL-15. Notably, the anti-PD-1-IL15 fusion showed improvement in anti-tumor activity and increased survival rates in mice compared to standalone Pembrolizumab treatment or co-treatment with IL-15-based molecule [19–22]. The fusion of Pembrolizumab with IL-15Rα-IL-15 complex was expected to release dual action by blocking PD-1 and PD-L1 interaction upon Pembrolizumab binding site and, in parallel, activating immune cell functions by IL-15 portion.

In this study, a fusion molecule was created by conjugating IL-15Rα-IL-15 complex to the C-terminus of Pembrolizumab using a flexible GGGS linker on both sides. The protein

assembly was produced using a plant molecular farming approach, where cloning plasmid was transformed into *Agrobacterium tumefaciens*, and bacteria vehicle was then infiltrated to *Nicotiana benthamiana* plants. This plant-based production system offers several advantages over the productions using insect and mammalian cells [23]. Plant-based approach is conceived as a green production system, free from animal uses. It provides a safe matrix, free of human pathogen contaminations [24]. However, plant-produced proteins would constitute a different *N*-glycan pattern from human glycans [25]. Therefore, *N*-glycans of Pembrolizumab-IL-15Rα-IL-15 were characterized using liquid chromatography-mass spectrometry (LC-MS) technique. Protein identity was also checked using sodium dodecyl sulfate-polyacrylamide gel electrophoresis (SDS-PAGE) and Western blot analyses. Finally, protein functions were tested using enzyme linked immunosorbent assay (ELISA) and mouse model for anti-tumor activity.

## Methods

### Research involving plants

The *Nicotiana benthamiana* seeds were obtained from Dr Supaart Sirikantaramas, Faculty of Science, Chulalongkorn University. Plant experiments were performed under controlled conditions at Faculty of Pharmaceutical Sciences, Chulalongkorn University with permission from the Institutional Biosafety Committee of Chulalongkorn University (CU-IBC). Plant handling and waste management was complied with the safety guidelines regulated by the Center for Safety, Health and Environment of Chulalongkorn University (SHECU).

### Approval for animal experiments

Mouse study was conducted at Gem Pharmatech Co., Ltd. (Nanjing, China) with approval by the Institutional Animal Care and Use Committee (IACUC). The care and use of animals was complied with the Animal Welfare Act and the Association for Assessment and Accreditation of Laboratory Animal Care (AAALAC). The approved study number was GPTAP20230823-6.

### Gene construction

Amino acid sequences of Pembrolizumab and Sushi domain of human IL-15 receptor subunit alpha (IL15Rα) and human IL-15 were retrieved from Drug Bank accession no. DB09037, UniProt accession no. Q13261 and P40933, respectively. The gene codons were in silico optimized using GeneArt Gene Synthesis program (Thermo Scientific, US). C-terminus of heavy chain of Pembrolizumab was ligated with IL15Rα using three sets of GGGGS and IL-15Rα was linked to IL-15 using four sets of GGGGS linker. The signal peptide of mouse IgG (UniProt accession no. P01750) was added to the N-terminus of both heavy chain and light chains of the constructs. The full sequence was synthesized using AccuOligo technology (Bioneer, South Korea). The graphic design of Pembrolizumab-IL-15Rα-IL-15 is depicted in **Fig 1** and the sequence details are described in **S1 Table**.

### Plasmid transformation

Nucleotide sequences of the heavy chain (HC) of pembrolizumab conjugated with IL15Rα-IL15 and the light chain (LC) of pembrolizumab were inserted into a pBAIYA geminiviral vector using XbaI and SacI restriction enzymes. The plasmids were transformed into *Agrobacterium tumefaciens* strain GV3101 using electroporation method. Transformed bacteria were spread on Luria-Bertani (LB) plate, supplemented with antibiotics, kanamycin, gentamycin and ampicillin (50 μg ml$^{-1}$ each) and grown at 28˚C for 48 h. Positive clone was grown for another 24 h in LB broth supplied with antibiotics and further used for infiltration. The

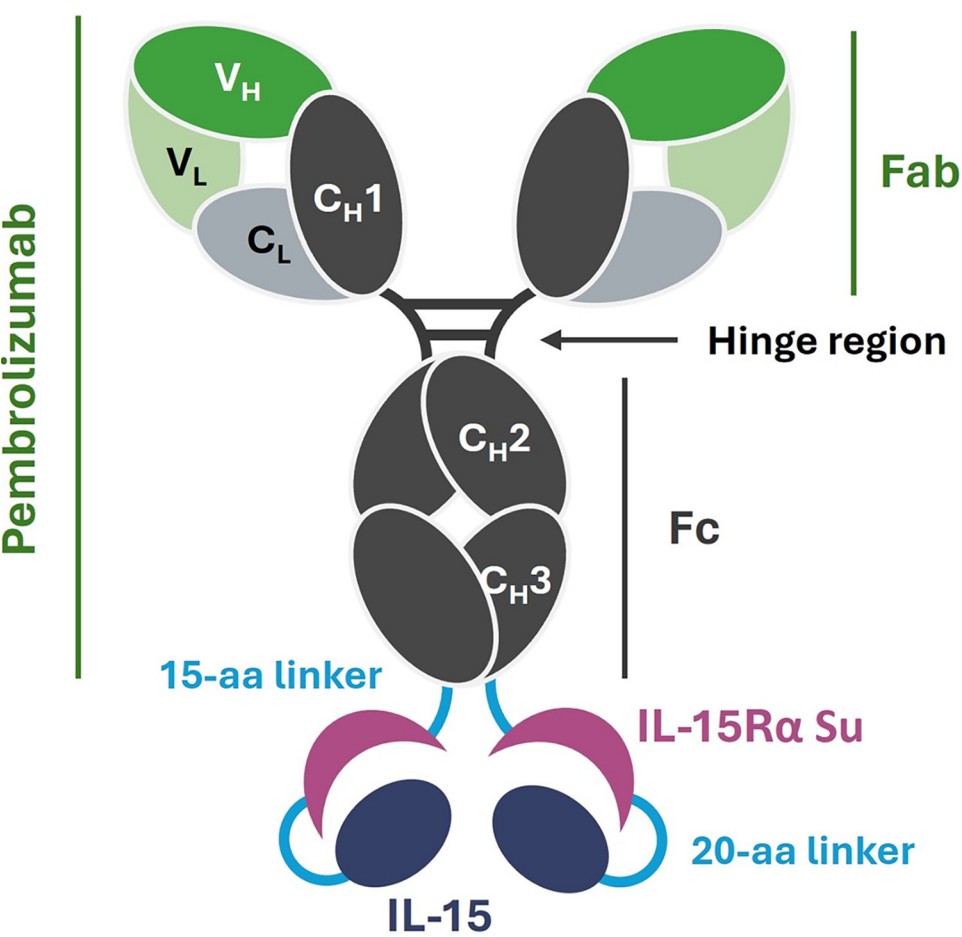

**Fig 1. Graphic structure of pembrolizumab-IL-15Rα-IL-15 molecule.** The main structure of pembrolizumab contains Fab region and Fc region. $V_H$; variable region of heavy chain, $C_H1$-3; constant region of heavy chain, $V_L$; variable region of light chain and $C_L$; constant region of light chain. Sushi domain of IL-15Rα is linked to both side of the C-terminal of pembrolizumab with a 15-aa linker (3 sets of GGGGS). IL-15 is linked to the C-terminal of IL-15Rα domain with a 20-aa linker (4 sets of GGGGS).

sequence of positive clones was confirmed by colony PCR and DNA sequencing. The colonies were diluted in 20 μl of water and used as a template for Taq polymerase amplification. The gel band was excised and purified using MEGAquick-spin plus DNA purification kit (iNtRON Biotechnology, Korea) and submitted for Sanger DNA sequencing using ABI 3730XL (Thermo Scientific, US).

### Transient expression

The infiltration step was conducted as described previously [26, 27]. Briefly, *Nicotiana benthamiana* was grown in hydroponic solution at room temperature, approximately 28˚C, with 16 h light and 8 h darkness. Three-week-old plants were infiltrated with both *A. tumefaciens* containing pembrolizumab-IL15Rα-IL15 and pembrolizumab LC using vacuum infiltration method. Infiltration buffer was 10 mM 2-(N-morpholino) ethanesulfonic acid (MES) and 10 mM MgSO$_4$, pH 5.5. After infection, plants were grown under the same growth condition for three days when wilting symptoms were observed.

## Extraction and purification

Plant leaves were collected and extracted using a blender in 1x phosphate buffer saline (PBS; 137 mM NaCl, 2.7 mM KCl, 4.3 mM $NaH_2PO_4$ and 1.47 mM $KH_2PO_4$, pH 7.4). Crude extract was filtered through filter cloth and centrifuged at 15,000 rpm, 4°C for 40 min. Supernatant was passed through a 0.45-μm membrane filter and the recombinant protein was purified using Protein A affinity column. The antibody was eluted using 100 mM glycine, pH 2.9 and neutralized with 1.5 M Tris buffer. The protein solution was dialyzed against 1x PBS cell grade for four cycles and concentrated with 50 kDa molecular weight cutoff (MWCO) Amicon Ultra-4 (Merck Millipore, US). Final protein concentration was measured using Bradford assay according to the manufacturer's protocol (Bio-Rad, US) and filtered through a 0.22-μm syringe filter. The purified protein was kept at -80°C for further analysis.

## SDS-PAGE and Western blot

SDS-PAGE analysis was carried out as described previously [28]. Briefly, crude extract along with flow-through, wash and eluate fractions were mixed with either reducing or non-reducing dyes and loaded into gels. Approximately 20 μg proteins were separated on 6% polyacrylamide gel and stained with Coomassie blue stain. The gel was transferred to a nitrocellulose membrane using Mini-Trans-Blot system (Bio-Rad, US). The membrane was blocked with 5% skim milk for 30 min and probed with goat anti-human gamma conjugated with horseradish peroxidase (HRP) at 1:10,000 dilution and goat anti-human kappa conjugated with HRP at 1:2,500 dilution for 2 h. The membrane was washed with 1x PBS and detected with enhanced chemiluminescence (ECL) substance.

## PD-1 binding assay

ELISA technique was used to determine binding activity of plant-produced pembrolizumab-IL-15Rα-IL-15 in comparison with commercial Keytruda. A 96-wells microtiter plate was coated with 2 μg ml$^{-1}$ of recombinant human PD-1 protein overnight. The plate was washed with 1x PBS-Tween (PBS-T) and blocked with 3% bovine serum albumin (BSA) for 2 h. Commercial Keytruda and plant-produced pembrolizumab-IL-15Rα-IL-15 were serially diluted and added to the plate with 2 h incubation at 37°C. Goat anti-human kappa-HRP at 1:3,000 dilution was used for probing and the peroxidase activity was developed by adding 3,3′,5,5′′-tetramethylbenzidine (TMB) substrate. The reaction was stopped using 1 M $H_2SO_4$ and measured at 450 nm using an EnSight multimode plate reader (PerkinElmer, US). Duplicate was performed per condition.

## LC-MS protein mass analysis

Approximately 20 μg protein was diluted in 50 mM ammonium bicarbonate (ABC) buffer. Protein solution was directly injected to LC-MS analysis for intact mass analysis and reduced with 10 mM dithiothreitol (DTT) at 65°C for 30 min for subunit mass analysis. The solution was centrifuged at 14,000 rpm for 10 min prior to the injection.

LC-MS was conducted using Agilent 1290 Infinity II LC system coupled with Agilent 6545XT Q-TOF mass spectrometer. LC separation was performed on Agilent PLRP-S column (1000 Å, 5 μm; 2.1 × 50 mm) at 60°C. Injection volume was 3 μl. Mobile phase A was 0.1% formic acid (FA) in water and mobile phase B was 0.1% FA in acetonitrile. LC gradient was set as follows; 25% at 0 min, 25% to 30% B in 1 min, 30% to 32% B in 2 min, 32% to 35% B in 1 min, 35% to 40% B in 1 min, 40% to 90% B in 1 min, 90% B for 3 min, 90% to 25% B in 0.5 min and 25% B for 2.5 min, with constant flow rate of 0.4 ml min$^{-1}$. MS analysis was conducted in

positive mode with a mass range of 100–3,200 m/z. MS parameters were set as follows; gas temperature at 350°C, nebulizer at 35 psi, dying gas at 12 L min$^{-1}$, sheath gas temperature at 400°C, sheath gas flow at 11 L min$^{-1}$, capillary voltage at 4,000 V, nozzle voltage at 2,000 V and skimmer voltage at 65 V. Acquisition time was 1 spectrum per s. Reference mass was monitored at 922.0098 m/z according to hexakis(1H, 1H, 3H-tetrafluoropropoxy)phosphazine (HP-0921) compound. Data was collected in profile mode.

## LC-MS peptide mapping

Protein solution with approximately 60 μg protein was reduced with 10 mM dithiothreitol (DTT) at 65°C for 30 min and alkylated with 25 mM iodoacetamide (IAA) at room temperature for 20 min under darkness. Proteins were digested with 0.6 μg trypsin at 37°C for 4 h. The reaction was stopped with 1% FA. The solution was then centrifuged at 14,000 rpm for 10 min and transferred to LC-MS vial.

Peptides were analyzed using Agilent 1290 Infinity II LC system coupled with Agilent 6545XT Q-TOF mass spectrometer. LC separation was conducted on AdvanceBio Peptide Mapping column (120 Å, 2.1 × 150 mm, 2.7 μm) at 60°C. Injection volume was 10 μl. Mobile phase A was 0.1% FA in water and mobile phase B was 0.1% FA in acetonitrile. LC gradient was set as follows; 0% B for 2 min, 0% to 20% B in 33 min, 20% to 30% B in 20 min, 30% to 50% B in 10 min, 50% to 90% B in 5 min, 90% B for 5 min, 90% to 0% B in 5 min and 0% B for 5 min, with constant flow rate of 0.4 ml min$^{-1}$. MS analysis was conducted in positive mode with MS and MS/MS mass range of 100–1,700 and 50–1,700 m/z, respectively. Acquisition times were 5 and 3 spectrum per s for MS and MS/MS, respectively. MS parameters were set as follows; gas temperature at 325°C, nebulizer at 35 psi, dying gas at 13 L min$^{-1}$, sheath gas temperature at 275°C, sheath gas flow at 12 L min$^{-1}$, capillary voltage at 4,000 V, nozzle voltage at 500 V and skimmer voltage at 65 V. Top 10 precursor ions per cycle were selected for MS/MS fragmentation. The absolute precursor threshold was 3,000 counts. Collision energy (CE) was varied according to the charge state of the peptide. For peptides with charge +1 and +2, the CE was calculated using a formula of $(3.1 \times ((m/z)/100) + 1)$, while peptides with charge $\geq +3$, the CE was calculated using a formula of $(3.6 \times ((m/z)/100) - 4.8)$. Reference mass was monitored at 922.0098 m/z. Data was collected in centroid mode.

## MS data analysis

LC-MS results were analyzed using Agilent MassHunter BioConfirm software version 11.0. For intact and subunit masses, protein peaks were integrated, and MS spectrum was extracted using default settings. Spectra with average scan > 10% of peak height were included. Spectrum was deconvoluted with a mass range of 100–250 kDa for intact mass and 20–100 kDa for subunit mass. Peak signal to noise was filtered with $\geq 30$ counts and minimum consecutive charge stage was committed at 5.

For peptide mapping, only peptides with MS/MS scan results were included. MS and MS/MS abundance filters were 1,000 and 50 counts, respectively. MS and MS/MS match tolerances were ±10 and ±30 ppm, respectively. False discovery rate (FDR) was set at 1%. Trypsin was selected as a digestion method. Cysteine carbamidomethylation, methionine oxidation and serine, threonine, tyrosine phosphorylation were selected as possible modifications.

## Anti-tumor study in mouse

The protocol for anti-tumor efficacy test was based on previous study [29]. Six-to-eight-week-old female C57BL/6-hPD1 mice were subjected to tumor induction. Mycoplasma-free MC38 colorectal cancer cells ($1\times10^{6}$ cells) were introduced subcutaneously to all mice. Mice were

maintained for nine days until the tumor grew to approximately 80 mm$^3$ in size. They were then divided into three groups: PBS vehicle, plant-produced pembrolizumab-IL-15Rα-IL-15 (3 mg kg$^{-1}$) and commercial Keytruda (5 mg kg$^{-1}$). Six mice were assessed per group. Drugs were administered intraperitoneally every 3 days (Q3D) for six doses. Tumor volume was measured every 2–3 days using the formula of $0.5 \times L \times W^2$, where L and W were tumor length and width, respectively. Mouse total body weight was also monitored until the last timepoint on day 27. At the study endpoint, mice were sacrificed by inhalation of 95% $CO_2$, and the tumors were subsequently collected and weighed.

## Results

### Design of Pembrolizumab-IL-15Rα-IL-15

The molecular structure of Pembrolizumab-IL-15Rα-IL-15 is illustrated in **Fig 1**. It contains three main parts: Pembrolizumab, IL-15Rα Sushi domain and human IL-15. Sushi domain is an extracellular domain of IL-15Rα that interacts with IL-15 cytokine and potentiates its functions. As compared to the designs of other studies, the molecule of this study contains a complex of IL-15Rα-IL-15 linked on both sides of Fc region. A flexible "GGGGS" linker was used to bridge Pembrolizumab with IL-15Rα and IL-15Rα with IL-15. The amino acid sequences of IL-15 cytokine were unmodified. The primary construct of Pembrolizumab-IL-15Rα-IL-15 is detailed in **S1 Table**.

### Protein production

*Agrobacterium tumefaciens* containing a plasmid of Pembrolizumab-IL-15Rα-IL-15 HC and one carrying a plasmid of Pembrolizumab LC were co-infiltrated in hydroponically grown *N. benthamiana* using a vacuum chamber. Three days after infiltration, plants demonstrated wilting symptoms and plant leaves were collected (**Fig 2A and 2B**). Two different *A. tumefaciens* clones (P1 and P2) were used for data comparison. Crude plant extracts showed similar band patterns on SDS-PAGE for both clones (**Fig 2C and 2D** and **S1 Fig**). Western blot analysis in non-reducing condition indicated a large protein band at above 250 kDa in the detections with both anti-human gamma and anti-human kappa antibodies (**Fig 2E and 2F** and **S2B-S2D Fig**). This band was likely a complete assembly of Pembrolizumab-IL-15Rα-IL-15 fusion protein. However, under reducing condition, protein band was not observed with anti-human gamma detection (**Fig 2G** and **S2A Fig**). This could be because the IL-15Rα-IL-15 fusion part could affect the binding pocket of Fc region that interacts with anti-gamma antibody. On the other hand, a faint protein band was observed at approximately 30 kDa with anti-human kappa antibody detection (**Fig 2H** and **S2C Fig**). This band was deemed as a LC chain of Pembrolizumab.

### Protein purification

After purification, one major protein band at above 250 kDa was detected from the non-reduced SDS-PAGE (**Fig 3A** and **S3A Fig**). This band was correlated to the band observed from the Western blot analysis (**Fig 2E and 2F**). Some smaller bands between 150–250 kDa were also detected, implying that some protein fragments could exist in the eluate after purification. Furthermore, SDS-PAGE was performed to check the capability of Pembrolizumab-IL-15Rα-IL-15 protein purification. It was also compared with the purification of unconjugated Pembrolizumab. The result showed that no corresponding protein band was found in the flow-through and wash fractions, demonstrating a competent binding between Pembrolizumab-IL-15Rα-IL-15 and protein A affinity column (**Fig 3B** and **S3B Fig**). In SDS-PAGE,

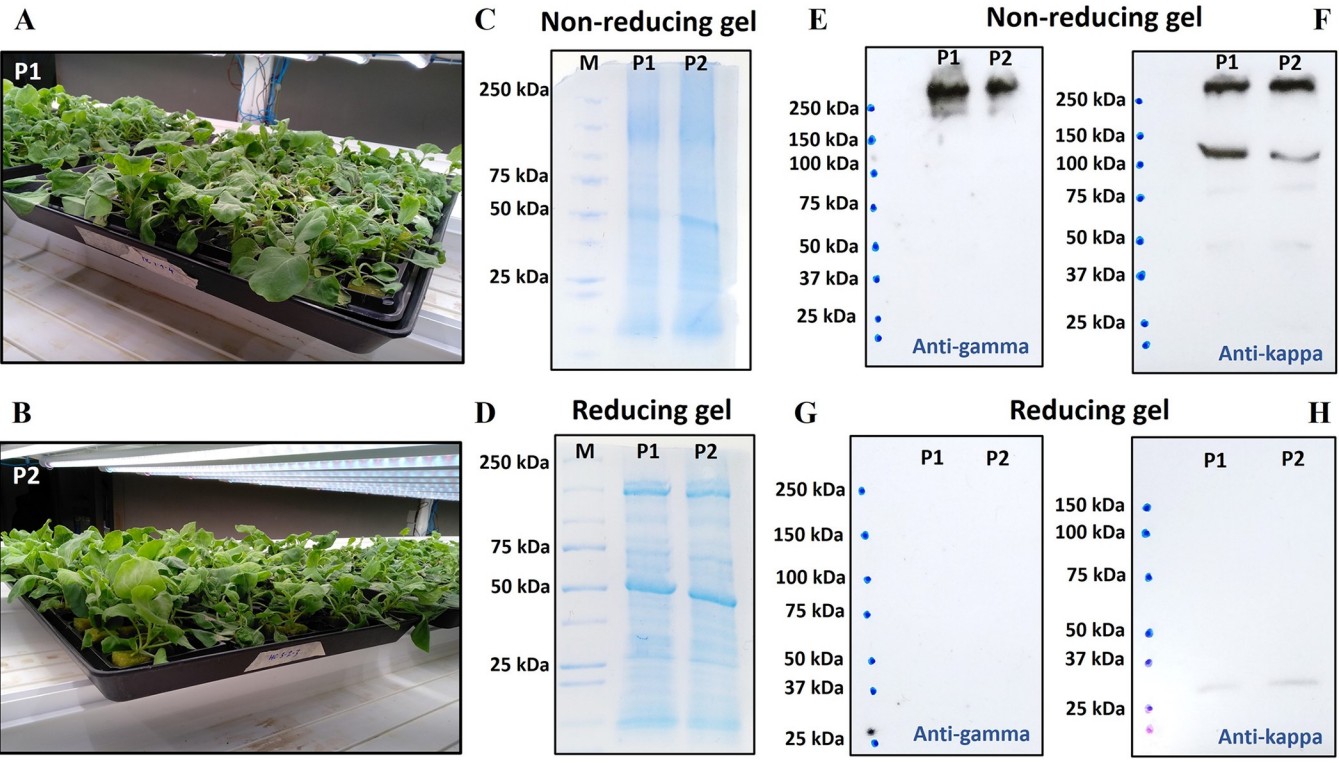

**Fig 2. Characterization of pembrolizumab-IL-15Rα-IL-15 fusion protein during protein production.** Plant morphology of *N. benthamiana* after three-day infiltration with *A. tumefaciens*, showing both clone P1 (A) and P2 (B). SDS-PAGE showing total proteins of crude *N. benthamiana* extracts under non-reducing (C) and reducing conditions (D). M is a Precision Plus Protein marker (Bio-Rad, US). Western blot analysis showing specific protein bands under non-reducing (E–F) and reducing conditions (G–H). The membrane was probed with anti-gamma (E and G) and anti-kappa (F and H) antibodies.

unconjugated Pembrolizumab band appeared at approximately 200 kDa in a non-reducing condition, suggesting that the protein band observed at above 250 kDa could be the Pembrolizumab-IL-15Rα-IL-15 fusion protein (**Fig 3B** and **S3B Fig**). After measuring protein concentration, protein yield was approximately 8.8 μg protein per 1 g leaf FW.

## PD-1 binding efficiency

As shown in **Fig 4** and **S2 Table**, the plant-produced Pembrolizumab-IL-15Rα-IL-15 showed comparable PD-1-binding pattern to commercial Keytruda. The 50% effective concentration (EC50) of Pembrolizumab-IL-15Rα-IL-15 (0.24 μg ml$^{-1}$) was approximately two times higher than that of Keytruda (0.12 μg ml$^{-1}$). However, statistical analysis revealed that the observed difference is not statistically significant (P-value >0.05), suggesting similar binding capacities between the two molecules under the different concentrations. Given that the same protein concentration was loaded onto the ELISA plate, the protein with a larger size could have lower number of molecules. Hence, higher concentration of Pembrolizumab-IL-15Rα-IL-15 would be required to equally bind to the PD-1 protein immobilized on the ELISA plate as compared to Keytruda. The result also suggests that the IL-15 fusion part at the C-terminus did not affect binding capacity of Pembrolizumab-IL-15Rα-IL-15 molecule to PD-1 protein.

## Protein primary structure

Based on LC-MS analysis, intact protein mass was 199.30 kDa (**Fig 5A and 5B**), which was 5.40 kDa higher than its theoretical mass of 193.66 kDa. This mass increase was potentially

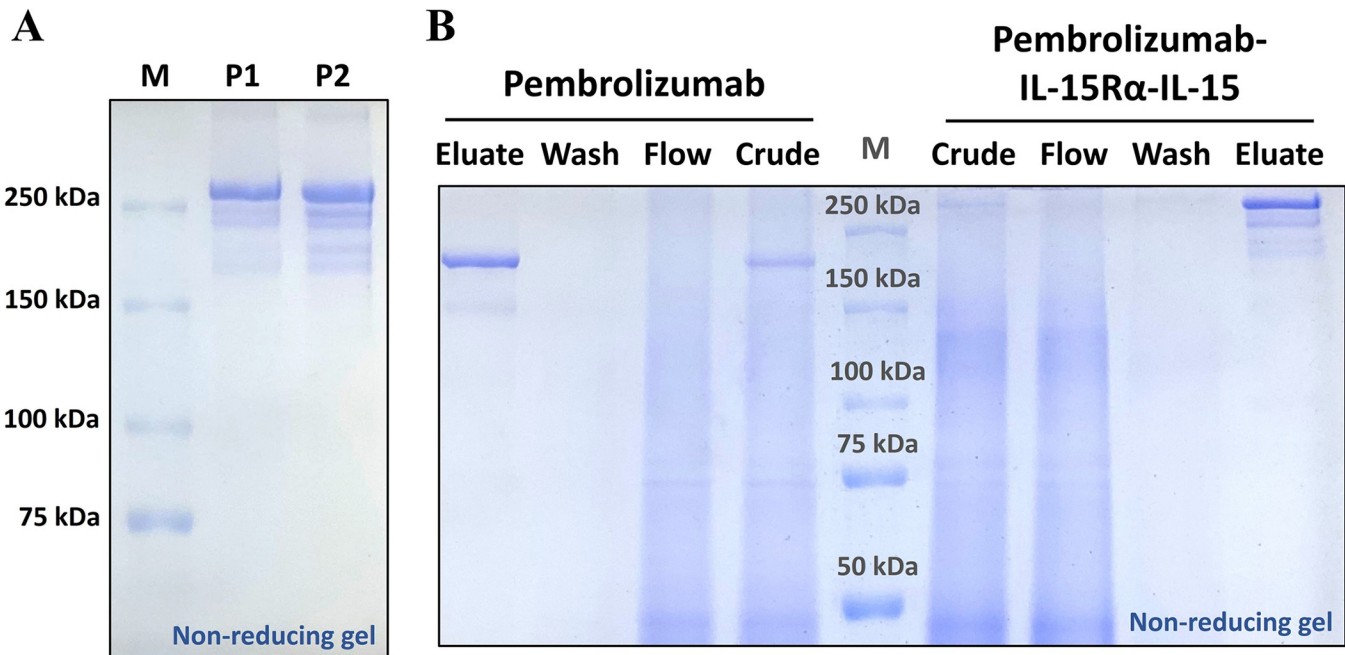

**Fig 3. Characterization of pembrolizumab-IL-15Rα-IL-15 fusion protein during protein purification.** SDS-PAGE showing eluted proteins of clone P1 and P2 after purified with protein A affinity column under non-reducing condition (A). SDS-PAGE of various protein fractions, including crude extract, flow-through, wash and eluate upon purification step of pembrolizumab and pembrolizumab-IL-15Rα-IL-15 under non-reducing condition (B). M is a Precision Plus Protein maker (Bio-Rad, US).

due to post-translational modifications, where *N*-glycosylation could be a leading factor owing to a large size of *N*-glycan unit with approximately 1.2–1.8 kDa. Subunit mass data confirmed the theoretical mass of LC chain, shown at 24.44 kDa (**S4A Fig**)**.** The experimental mass of heavy chain was 73.32 kDa, which was 0.9 kDa-larger than its theoretical mass of 72.41 kDa (**S4B Fig**). *N*-glycosylation modifications and protein truncations could be primary factors contributing to the mass shift.

In peptide mapping analysis, a total of 2,751 peptides were detected (**Fig 5C**). Among them, 603 peptides were matched to the reference sequence of Pembrolizumab-IL-15Rα-IL-15, resulting in 99.66% sequence coverage (**Fig 5D**). Peptides corresponding to the IL-15 fusion parts and the linkers were all observed (**Fig 5D**), confirming complete attachment of the IL-15Rα-IL-15 complex to the C-terminus of Pembrolizumab. Post-translational modifications (PTMs) were also detected at various positions of the molecule, such as oxidations at M111, M258 and M475 and phosphorylation at S604 (**S3 Table**). *N*-glycosylation was a significant PTMs of molecular-mass changes. From MS data, different glycans were detected at N303 position, a common glycosylated site of Fc region. The corresponding peptide of EEQFNSTYR were found attached with different plant glycans, including GnGn, GnGnXF, Man7GlcNAc2, Man8GlcNAc2 and Man9GlcNAc2 (**Fig 6A and 6B**). As shown in **Fig 6A** chromatogram, peptides attached with high-mannose glycans were eluted at around 14.8–14.9 min, faster than the peptides conjugated with complex glycans (15.1–15.6 min) and non-glycosylated peptides (16.1 min), respectively. Around 45% of the molecule was glycosylated (**Fig 6B**). On the other hand, *N*-glycosylation at the other positions of the molecule were barely observed. The highly possible glycosylated site of IL-15 portion was reported at N632 position. While N624 and N665 positions, other possible glycosylated sites based on the NXT and NXS sequences, are unlikely to be glycosylated due to the quaternary configuration of the molecule [30]. Based on

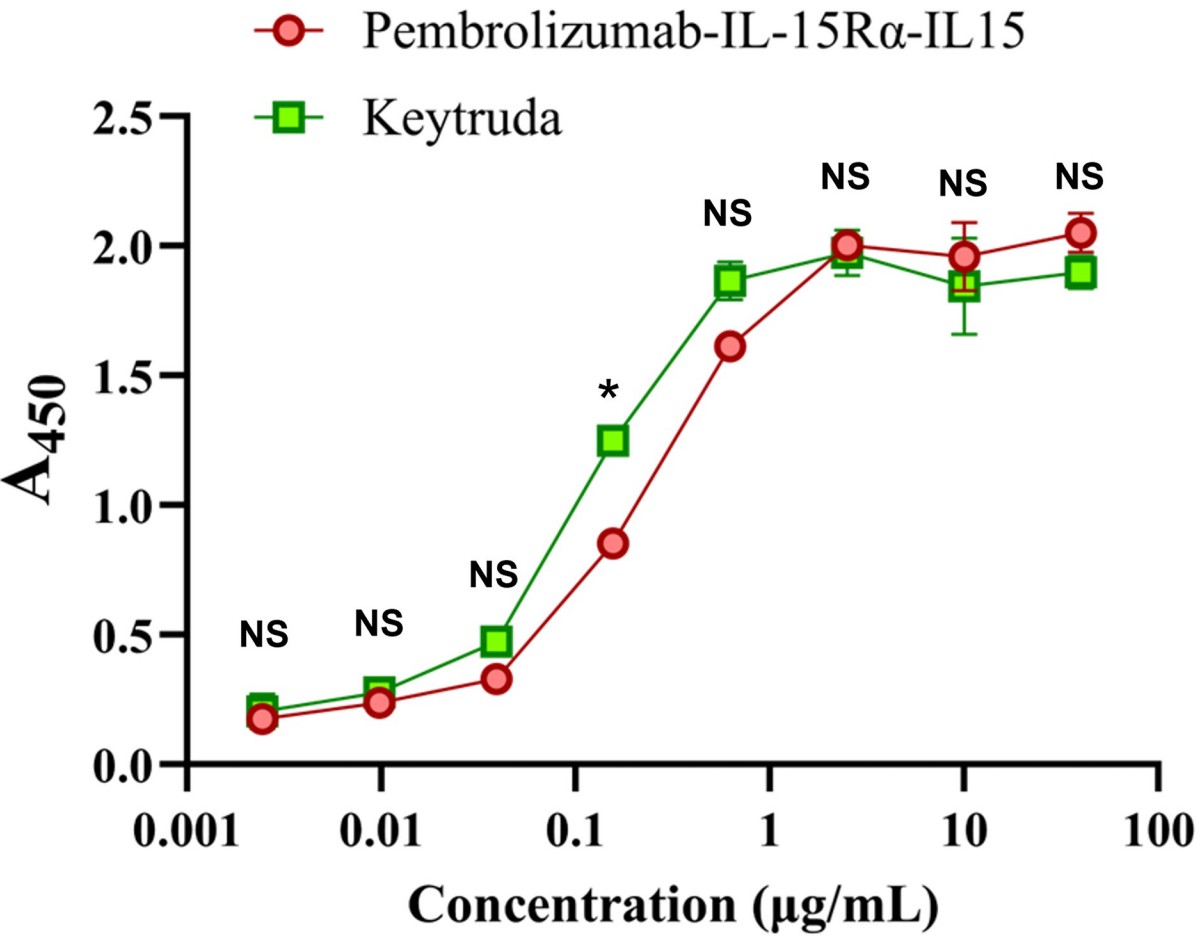

| Name | EC50 (µg/mL) | R² |
|---|---|---|
| Pembrolizumab-IL-15Rα-IL-15 | 0.242718 | 0.996051 |
| Keytruda | 0.118751 | 0.989790 |

**Fig 4. PD-1 binding activity of plant-produced Pembrolizumab-IL-15Rα-IL-15 in comparison with Keytruda using ELISA.** Eight concentrations were applied for the assay. * p-value < 0.05, NS: not significant difference and EC50: effective concentration showing 50% binding properties to PD-1 protein.

peptide mapping results, peak intensities of the glycosylated peptides were only 5.55% of the non-glycosylated peak (**S5 Fig**). The glycans observed were GnXF and GnGnXF forms.

## Anti-tumor activity

The study design of anti-tumor test in mouse model is shown in **Fig 7A**. The dose of Pembrolizumab-IL-15Rα-IL-15 was applied at 3 mg kg$^{-1}$, while Keytruda as a positive control was applied with a higher dose of 5 mg kg$^{-1}$. After 27 days of the first treatment, plant-produced Pembrolizumab-IL-15Rα-IL-15 significantly inhibited colon cancer growth with an average

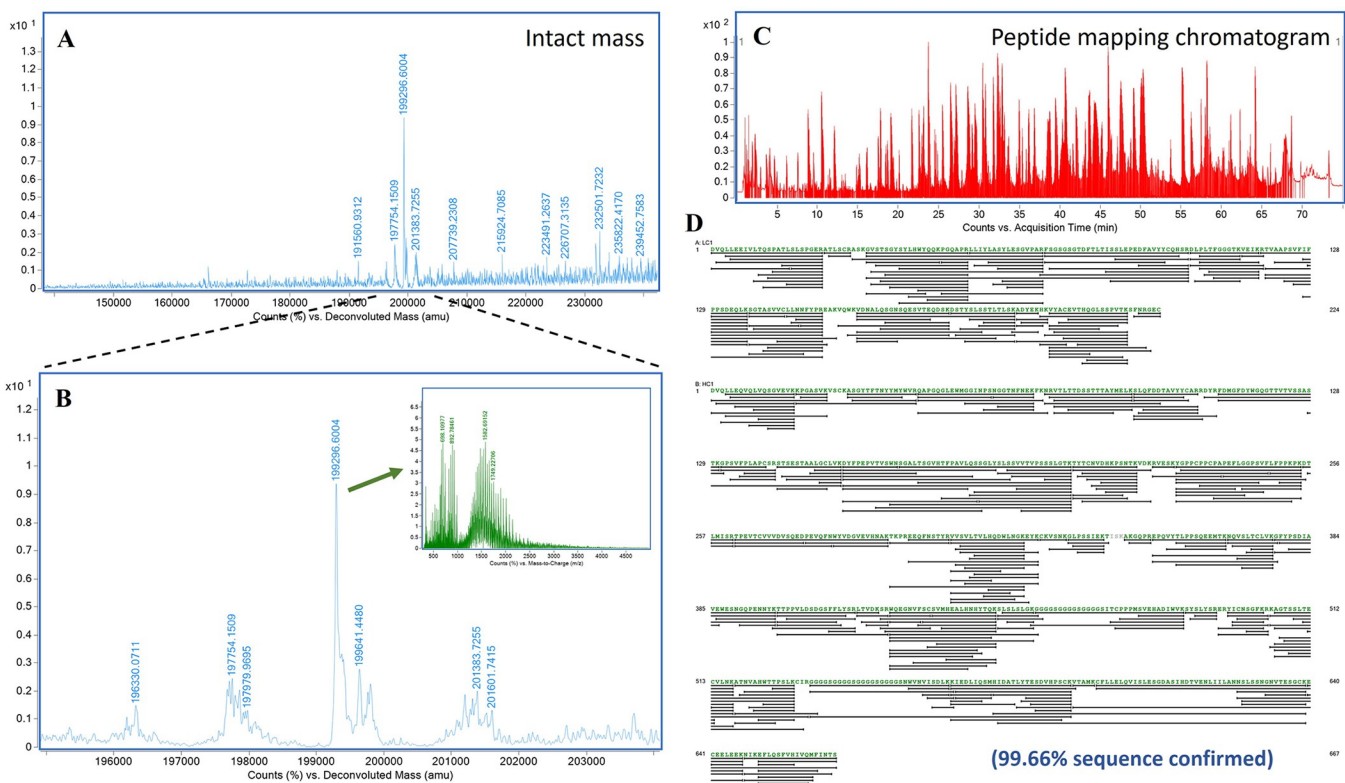

**Fig 5. Protein mass analysis of pembrolizumab-IL-15Rα-IL-15 using LC-MS.** Deconvoluted spectrum of intact protein showing major peak at 199.30 kDa (A). Zoomed-in spectrum showing an inset of raw MS spectrum (B). Peptide mapping chromatogram featuring 2,751 peaks detected in LC-MS (C). Map of 603 peptides detected with MS/MS fragments and matched to the reference sequence of pembrolizumab-IL-15Rα-IL-15 (D). The total match was summated to 99.66% sequence coverage.

final tumor volume of 103.11 ± 85.68 mm$^3$. It was 94.98% decreased from that of the vehicle group (1,933.72 ± 188.69 mm$^3$). The Keytruda group also had a significantly smaller final tumor volume with 86.55% reduced from the control (**Fig 7B and 7C**). This result suggested that plant-produced Pembrolizumab-IL-15Rα-IL-15 at the lower dose of 3 mg kg$^{-1}$ inhibited tumor growth at a comparable level to 5 mg kg$^{-1}$ of Keytruda. Likewise, the result of tumor weight (**Fig 7D and 7E**) was correlated with the result of tumor volume, where final tumor weights of the mice treated with Pembrolizumab-IL-15Rα-IL-15 were 94.14% decreased (0.10 ± 0.09 g) and the mice treated with Keytruda were 83.43% decreased (0.29 ± 0.19 g) from the control group (1.77 ± 0.13 g). Remarkably, four out of six mice treated with plant-produced Pembrolizumab-IL-15Rα-IL-15 were free of tumor at the collection point, day 27 (**Fig 7E**). The remaining two had relatively smaller tumor size than the mice treated with PBS vehicle. Nonetheless, final tumor weights between plant-produced Pembrolizumab-IL-15Rα-IL-15 and Keytruda groups were not significantly different (P = 0.31).

In addition, mouse body weights were not significantly different among three groups along the study (**Fig 7F**). Mouse body weight was 19.6–19.9 g on day 0 and reached 22.6–22.9 g on day 27. Pembrolizumab-IL-15Rα-IL-15 slightly affected mouse body weight during the first two weeks of the treatment. The average mouse body weight of the control group on day 14 was 21.6 g, 10.59% increased from its initial body weight. The average mouse weight of the Pembrolizumab-IL-15Rα-IL-15 group was only 20.4 g, 3.33% increased from its initial body weight, implying that mice did not grow normally at the start of the treatment. Nonetheless,

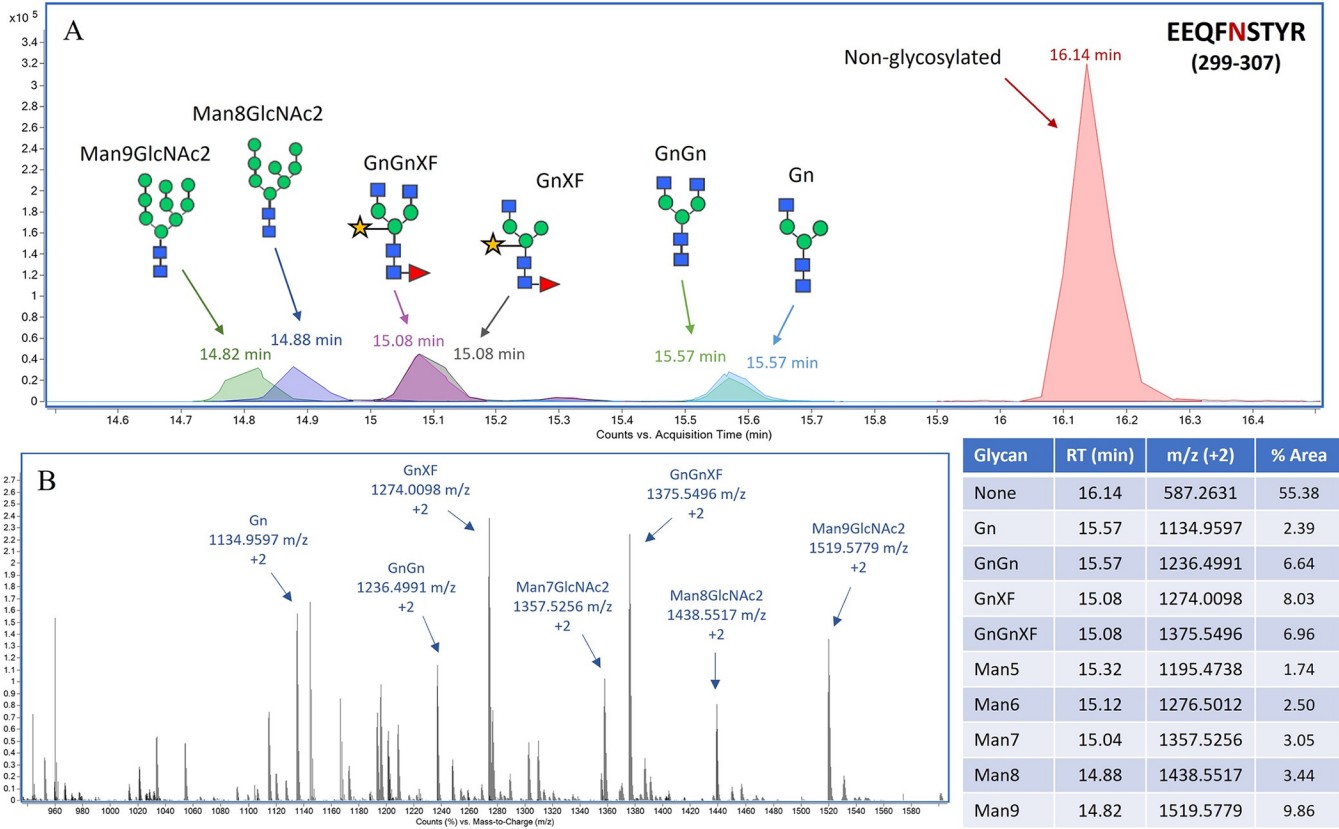

**Fig 6. *N*-glycosylation analysis at N303 position of pembrolizumab-IL-15Rα-IL-15 heavy chain.** Chromatogram of EEQFNSTYR peptide showing peaks of different plant *N*-glycan attachments (A). Non-glycosylated peak was eluted at 16.1 min, while glycosylated peaks were eluted faster between 14.8–15.6 min. MS spectrum of glycosylated mass from 950–1,600 m/z (B). Inset table summarizing *N*-glycosylation data, where 55.4% of the molecule was not glycosylated. Among 45.6% glycosylated peptides, GnXF, GnGnXF and Man9GlcNAc2 were major forms of attached *N*-glycans.

mouse body weight of the Pembrolizumab-IL-15Rα-IL-15 group ended at 14.20% growth from the initial size. This was comparable to the final weight of mice in the control and Keytruda groups, showing 16.53% and 15.27% growth, respectively.

## Discussion

Monoclonal antibodies (mAbs)-cytokine fusions have been developed to promote efficacy, reduce adverse effects and prolong compound longevity [31, 32]. A fusion between Pembrolizumab and IL-15 cytokine is deemed as a prominent anti-cancer molecule since Pembrolizumab mAb has shown clinical success in treating various cancers and a number of pre-clinical studies has underlined immunostimulatory effects of IL-15 cytokine [12, 33]. Several studies have modified the sequence of IL-15 to improve its efficacy and reduce side effects and combined it with IL-15Rα to form an active complex, readily activating immune cell generation and proliferation [34, 35]. This IL-15Rα-IL15 conjugation was expected to facilitate T cells and NK cells activation, without a primary formation of IL-15 with IL-15Rα on the antigen-presenting cells. In conjugation with Pembrolizumab, the molecule was deemed to enhance anti-tumor activity and reduce adverse effects of unconjugated cytokine payload.

In this study, pembrolizumab-IL-15Rα-IL-15 fusion was designed and produced using a plant molecular farming approach. The amino acid sequence of IL-15 cytokine was not modified, maintaining its natural structure. As proof of concept, this study has demonstrated that

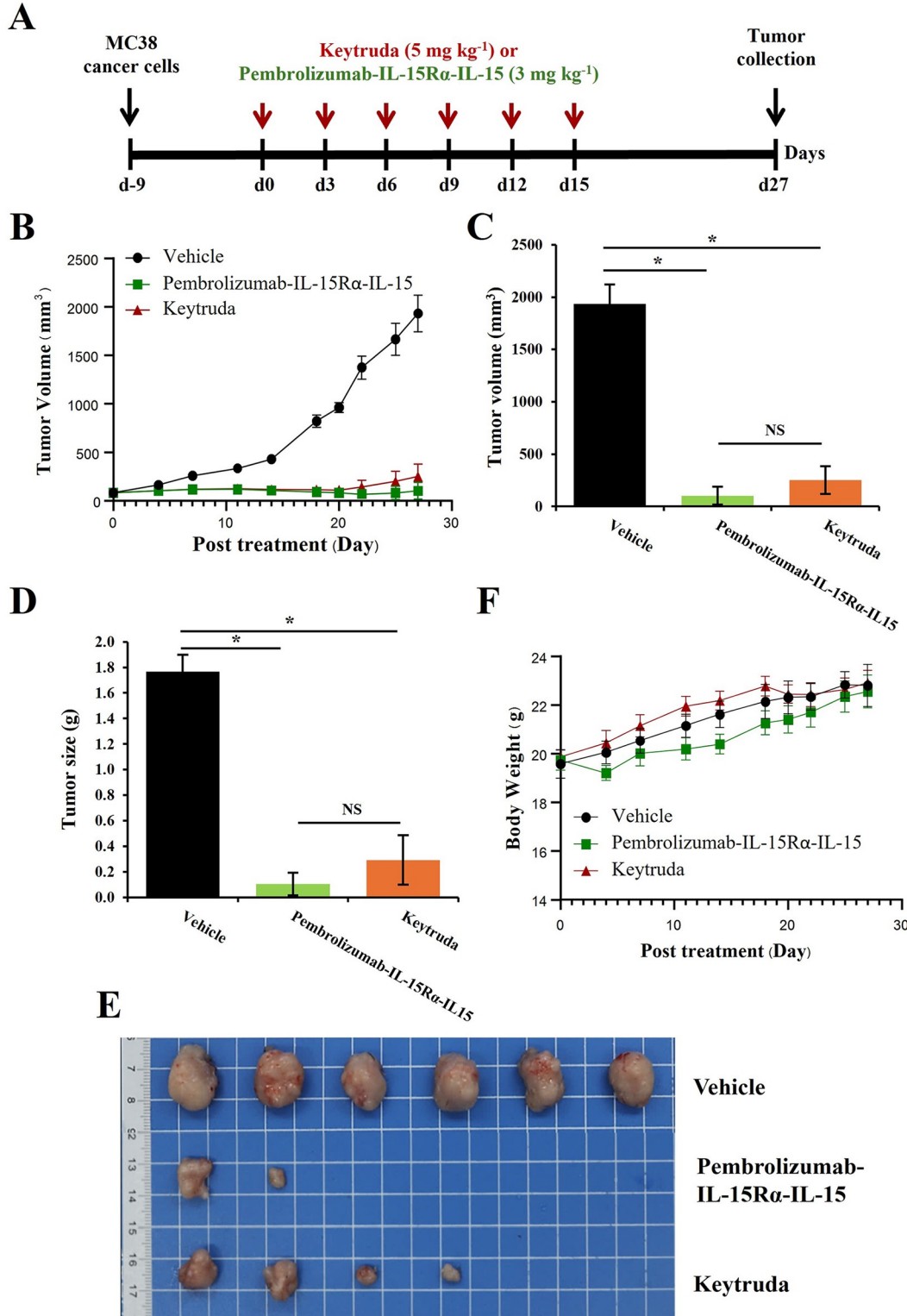

**Fig 7. Anti-tumor activity of pembrolizumab-IL-15Rα-IL-15 as compared to commercial Keytruda.** (A) The study design, showing MC38 cancer-cell inoculation nine days before the first treatment. Pembrolizumab-IL-15Rα-IL-15 and Keytruda were applied every three days (Q3D) for six doses and mice were maintained until day 27. (B) Tumor volume measured from day 0 till day 27 among three groups: vehicle control, pembrolizumab-IL-15Rα-IL-15 (3 mg kg$^{-1}$) and Keytruda (5 mg kg$^{-1}$). Bar graphs comparing final tumor volume (C) and tumor size (D) on day 27 among three groups. (E)

Morphological comparison of final tumor sizes on day 27 among three groups (n = 6). (F) Mouse body weight measured from day 0 till day 27 among three groups. Asterisk (*) indicated significant difference (P < 0.05) between two groups using T-test. NS is referred to non-significant difference.

plant molecular farming is an applicative platform to produce human immunocytokine compounds. The molecule of pembrolizumab-IL-15Rα-IL-15 is more complex than typical mAbs, e.g., pembrolizumab, atezolizumab and nivolumab, which their productions using plant molecular farming approaches were thoroughly displayed in previous studies [26, 28, 36]. As shown in the results, the production of pembrolizumab-IL-15Rα-IL-15 was accomplished using a typical plant molecular farming workflow, similar to the production of those mAbs. A wild-type *N. benthamiana* plant was applied along with basic agroinfiltration method using a vacuum chamber. Plants were maintained for three days after infection. However, the production yield of pembrolizumab-IL-15Rα-IL-15 was considerably lower than the productions of those unconjugated mAbs. Only 8.8 µg protein per g leaf FW was observed in this study, while the yields of pembrolizumab, nivolumab and atezolizumab were 4–130, 70–140 and 500–1,800 µg per g leaf FW, respectively [26, 28, 36].

The cultivation system was a key distinctive factor, where plants were grown under hydroponic system in this study, while soil-based systems were applied in previous setups. There is no clear evidence demonstrating that hydroponic system would lessen the yield of recombinant protein production, but during growth in hydroponic, plants were reported to secrete recombinant protein into root exudate [37]. A comparative study is therefore required to verify the effect of hydroponic platform in mAb production in comparison with the use of soil pot system. To improve production yield, several methods could be further implemented for Pembrolizumab-IL-15Rα-IL-15 production. For example, prolonging the day after infection period from 3 days to 4–6 days would allow plants to produce more proteins. This is because production of Pembrolizumab peaked at day 4 [26] and the yields of atezolizumab and nivolumab production were highest on day 5 and 6 according to previous studies [28, 36]. Adjusting nutrient solution and controlling plant stresses during infection would leverage plant productivity [38]. As previously reported, challenging plant leaves with antioxidant compounds, such as ascorbic acid and lipoic acid, can alleviate the levels of reactive oxygen species during *A. tumefaciens* infection and promote bacteria transformation, resulting in higher protein yield. In addition, surfactants, such as Tween-20 and Triton-X, could aid bacteria penetration according to their chemical properties to reduce leaf surface tension [25]. The downstream purification process could be another factor affecting production yield. Protein conformation of Pembrolizumab-IL-15Rα-IL-15 would be different from typical mAbs due to the additional part of IL-15 complex. Protein purification using protein A affinity column might not fully capture the fusion protein due to the hindrance of IL-15 fusion part on the protein A-binding pockets of Pembrolizumab Fc region [39]. In addition, suboptimal purification conditions, such as weak buffer system and too high or low pH, could diminish capability of protein A binding and impair protein stability upon the elution process [40].

In part of protein characterization, SDS-PAGE, Western blot and LC-MS analyses were applied to confirm Pembrolizumab-IL-15Rα-IL-15 identity. Under non-reducing condition of SDS-PAGE and Western blot analyses, protein band of Pembrolizumab-IL-15Rα-IL-15 was observed at above 250 kDa as expected. It was larger than the band of the unmodified Pembrolizumab detected at approximately 200 kDa. The size of IL-15Rα-IL-15 fusion part is approximately 22.3 kDa and the attachments of IL-15Rα-IL-15 on both side of Fc region of Pembrolizumab could cause such an increase to the protein size. In addition, smaller bands observed at approximately 200–250 kDa on the SDS-PAGE gel were likely partial or cleaved

pembrolizumab heavy chains. They might be co-purified with a complete form of Pembrolizumab-IL-15Rα-IL-15 by protein A affinity column. After purification, these heavy chain fragments appeared in the drug product and were detected with Coomassie blue protein staining due to protein being. However, they were not detected in the Western blot analysis because of an incomplete antibody formation by heavy chain and light chain, resulting in lack of binding epitope for anti-Gamma and anti-Kappa antibodies to detect.

In LC-MS analysis, peptide mapping analysis confirmed the protein sequence of Pembrolizumab-IL-15Rα-IL-15, where 99.66% of amino acids were identified with MS/MS data. The intact protein mass of Pembrolizumab-IL-15Rα-IL-15 was detected at 199.30 kDa, larger than its theoretical mass of 193.66 kDa. This increase was likely due to *N*-glycan additions to the Fc region of Pembrolizumab. Pembrolizumab-IL-15Rα-IL-15 protein contains two possible *N*-glycosylated sites–one at N303 position located on Fc region of Pembrolizumab and another one at N632 position located on IL-15 portion. Approximately 45% of N303 position was glycosylated with various plant glycans. The major forms of detected glycan were GnGnXF, GnXF, Man8GlcNAc2 and Man9GlcNAc2, common *N*-glycans found in plants [41]. On the other hand, approximately 95% of N632 position was not glycosylated, suggesting that only Fc region of Pembrolizumab-IL-15Rα-IL-15 fusion protein was glycosylated. This was in contrast with previous study indicating that 95% of IL-15 part of IL-15-sIL-15Rα fusion complex was glycosylated with G0F and G0FN human glycans [42]. This could be because the protein sequence and production strategy of IL-15/sIL-15Rα fusion was different from the Pembrolizumab-IL-15Rα-IL-15 fusion protein of this study. The heterodimeric IL-15-sIL-15Rα was produced using HEK293 human cell lines and the protein complex was not conjugated with mAb. Additionally, the sequence of sIL-15Rα included full-length human IL15R soluble part, which was different from the specific extracellular domain of IL-15R of the current Pembrolizumab-IL-15Rα-IL-15 molecule. Nevertheless, further validation of *N*-glycosylated sites and glycan patterns of Pembrolizumab-IL-15Rα-IL-15 molecule would be required to confirm the final protein product element. The LC-MS-based method of released *N*-glycan profiling could be a useful analysis, providing informative data [42]. Variation in *N*-glycosylation profiles would influence binding capability of the molecule to IL-2Rβ/γ and Fc receptors in human body, leading to changes in compound efficacy, side effects and half-life [43].

In terms of activity tests, Pembrolizumab-IL-15Rα-IL-15 bound to PD-1 protein in a similar binding pattern to the unmodified Pembrolizumab. However, double concentration of Pembrolizumab-IL-15Rα-IL-15 molecule was required to bind with coated PD-1 protein on the ELISA plate. This was likely because of a larger size of Pembrolizumab-IL-15Rα-IL-15 than Pembrolizumab, resulting in a lower number of Pembrolizumab-IL-15Rα-IL-15 molecules as compared to neat Pembrolizumab at equivalent concentration. To further validate compound binding efficacy, binding capacity toward IL-2Rβ and IL-2Rγ and competitive binding to PD-L1 protein should be examined. In addition, competitive binding to PD-1 protein between unconjugated Pembrolizumab and Pembrolizumab-IL-15Rα-IL-15 should be further carried out to directly compare PD-1 binding efficacy of both molecules side-by-side.

In mouse study, Pembrolizumab-IL-15Rα-IL-15 expressed acceptable tumor growth inhibition. It inhibited tumor growth at a comparable level to a commercial Keytruda with slight improvement. At low dose of 3 mg kg$^{-1}$, four out of six mice treated with Pembrolizumab-IL-15Rα-IL-15 were free of tumor development, while tumor sizes of the other two mice were much smaller than that of control. Whereas only two out of six mice treated with 5 mg kg$^{-1}$ of Keytruda were free of tumor growth. This result was correlated with previous studies, reporting that anti-PD-1-IL15 fusion proteins increasingly suppressed tumor growth and extended mouse survival rate as compared to the unmodified anti-PD-1 drugs [20, 22]. Those studies also showed consistent results among different cancer models, including colorectal cancer,

pancreatic cancer and melanoma. Improved anti-tumor effect of fusion proteins could be due to a dual mode of actions, rendered by both monoclonal antibody and therapeutic cytokine [44].

Furthermore, the body weight of mice treated with Pembrolizumab-IL-15Rα-IL-15 was slightly lower than that of control and Keytruda groups during the first two weeks. However, it increased to normality over the last two weeks. This result indicates that the Pembrolizumab-IL-15Rα-IL-15 fusion protein would cause acute weight loss. This phenomenon was also observed in previous study, noticing significant weight loss during day 5–9 after the treatment with anti-PD-1-IL-15 mutant at 5 mg kg$^{-1}$ [20]. Nonetheless, a decreased dose of anti-PD-1-IL-15 mutant at 1 mg kg$^{-1}$ could maintain compound efficacy but did not cause weight loss. With this regard, toxicity profiles of plant-produced Pembrolizumab-IL-15Rα-IL-15 will be further examined using immunogenicity test along with monitoring long-term pharmacokinetic parameters with dose response assay to ensure compound efficacy and safety across varied dose at different timepoints [19]. Also, efficacy tests will be further investigated in different cancer models. These further investigations will provide strong evidence to support the development and potential use of this fusion molecule.

Overall, plant-produced Pembrolizumab-IL-15Rα-IL-15 fusion protein showed effective anti-tumor activity in mouse models induced with colorectal tumor. It might not be superior to the commercial Pembrolizumab but showed signs of improvement. Further pre-clinical and clinical studies are required to examine its potency and safety using dose response in different cancer models. The design of Pembrolizumab-IL-15Rα-IL-15 fusion protein could be a prototype for developing an improved version of immunoregulatory substances.

## Conclusion

A Pembrolizumab-IL-15Rα-IL-15 fusion molecule was successfully developed and produced using a plant molecular-based approach. The results of protein characterization indicated successful protein fusion, expression and purification. The molecule exhibited comparable anti-cancer activity in PD-L1-humanized mice as commercial Keytruda. It could be a potential compound for cancer immunotherapy and an important prototype for immunotherapeutic drug development, where a plant molecular farming could be a valuable platform for protein manufactures. Nonetheless, the studies of toxicity and pharmacokinetic profile are necessary to fulfil pharmacological background of plant-produced Pembrolizumab-IL-15Rα-IL15.

## Supporting information

**S1 Table. Details of pembrolizumab-IL-15Rα-IL-15 sequence.**
(DOCX)

**S2 Table. Mean absorbance of PD-1 binding data and statistical comparison between Pembrolizumab-IL-15Rα-IL15 versus Keytruda.**
(DOCX)

**S3 Table. Post-translational modifications of pembrolizumab-IL-15Rα-IL-15 detected with LC-MS peptide mapping analysis.**
(DOCX)

**S1 Fig. Raw files of SDS-PAGE gels shown in Fig 2C and 2D.** The cropped areas are indicated in red.
(DOCX)

**S2 Fig. Raw file of Western blot films shown in Fig 2E–2H.** The cropped areas are indicated in red.
(DOCX)

**S3 Fig. Raw files of SDS-PAGE gels shown in Fig 3A, 3B.** The cropped areas are indicated in red.
(DOCX)

**S4 Fig. Subunit mass analysis of pembrolizumab-IL-15Rα-IL-15 using LC-MS.** (A) Deconvulated peak of pembrolizumab light chain. Protein peak was observed at 24.4 kDa with an inset showing MS spectrum from 500–3,000 m/z. (B) Deconvulated peak of pembrolizumab-IL-15Rα-IL-15 heavy chain. Protein peak was observed at 73.3 kDa with an inset showing MS spectrum from 800–3,000 m/z.
(DOCX)

**S5 Fig. *N*-glycosylation analysis at N632 position of pembrolizumab-IL-15Rα-IL-15 heavy chain.** Chromatogram of specific peptide showing non-glycosylated and GnGnXF-glycosylated peaks (A). Non-glycosylated peak was eluted at 60.9 min, while glycosylated peaks were eluted at 60.2 min. MS spectrum of GnXF and GnGnXF glycosylated mass from 1490–1600 m/z (B). Inset table summarizing *N*-glycosylation data, where 94.45% of the molecule was not glycosylated.
(DOCX)

**S1 Raw images.**
(PDF)

## Author Contributions

**Conceptualization:** Pipob Suwanchaikasem, Waranyoo Phoolcharoen.

**Formal analysis:** Kaewta Rattanapisit, Pipob Suwanchaikasem, Christine Joy I. Bulaon, Shiying Guo.

**Funding acquisition:** Waranyoo Phoolcharoen.

**Investigation:** Kaewta Rattanapisit, Pipob Suwanchaikasem, Christine Joy I. Bulaon, Shiying Guo.

**Methodology:** Kaewta Rattanapisit, Pipob Suwanchaikasem, Christine Joy I. Bulaon, Shiying Guo.

**Supervision:** Waranyoo Phoolcharoen.

**Writing – original draft:** Pipob Suwanchaikasem, Christine Joy I. Bulaon.

**Writing – review & editing:** Kaewta Rattanapisit, Pipob Suwanchaikasem, Christine Joy I. Bulaon, Shiying Guo, Waranyoo Phoolcharoen.

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
