## [Decision Letter · Decision Letter 0]

29 Nov 2024

PONE-D-24-40114Plant-derived Pembrolizumab in conjugation with IL-15Rα-IL-15 complex shows improvement of anti-tumor activityPLOS ONE

Dear Dr. Suwanchaikasem,

Thank you for submitting your manuscript to PLOS ONE. After careful consideration, we feel that it has merit but does not fully meet PLOS ONE’s publication criteria as it currently stands. Therefore, we invite you to submit a revised version of the manuscript that addresses the points raised during the review process.

Please modify your manuscript according to the suggestions brought forward by the reviewers. Please discuss the reasons where this might not be justifyable.

We look forward to receiving your revised manuscript.

Kind regards,

Michael C Burger, M.D.

Academic Editor

PLOS ONE

Journal Requirements: When submitting your revision, we need you to address these additional requirements. 1. Please ensure that your manuscript meets PLOS ONE's style requirements, including those for file naming. The PLOS ONE style templates can be found at https://journals.plos.org/plosone/s/file?id=wjVg/PLOSOne_formatting_sample_main_body.pdf and https://journals.plos.org/plosone/s/file?id=ba62/PLOSOne_formatting_sample_title_authors_affiliations.pdf 2. Thank you for stating the following financial disclosure: "National Research Council of Thailand (NRCT) and Chulalongkorn University (Grant number: N42A670577) and the Thailand Science Research and Innovation Fund Chulalongkorn University (Grant number: BCG66330001) K.R. was supported by the National Science, Research, and Innovation Fund (NSRF) via the Program Management Unit for Human Resources & Institutional Development, Research, and Innovation (Grant number: B13F660137)" Please state what role the funders took in the study.  If the funders had no role, please state: ""The funders had no role in study design, data collection and analysis, decision to publish, or preparation of the manuscript."" If this statement is not correct you must amend it as needed. Please include this amended Role of Funder statement in your cover letter; we will change the online submission form on your behalf. 3. Thank you for stating the following in the Competing Interests section: "W.P. is a co-founder of Baiya Phytopharm Co., Ltd., Thailand. K.R., P.S., and C.J.I.B. are employees of Baiya Phytopharm Co., Ltd.  S.G. is an employee of GemPharmatech Co., Ltd. The remaining author has no competing financial interests. There are no relevant non-financial interests to declare." Please confirm that this does not alter your adherence to all PLOS ONE policies on sharing data and materials, by including the following statement: ""This does not alter our adherence to  PLOS ONE policies on sharing data and materials.” (as detailed online in our guide for authors http://journals.plos.org/plosone/s/competing-interests).  If there are restrictions on sharing of data and/or materials, please state these. Please note that we cannot proceed with consideration of your article until this information has been declared.  Please include your updated Competing Interests statement in your cover letter; we will change the online submission form on your behalf. 4. PLOS ONE now requires that authors provide the original uncropped and unadjusted images underlying all blot or gel results reported in a submission’s figures or Supporting Information files. This policy and the journal’s other requirements for blot/gel reporting and figure preparation are described in detail at https://journals.plos.org/plosone/s/figures#loc-blot-and-gel-reporting-requirements and https://journals.plos.org/plosone/s/figures#loc-preparing-figures-from-image-files. When you submit your revised manuscript, please ensure that your figures adhere fully to these guidelines and provide the original underlying images for all blot or gel data reported in your submission. See the following link for instructions on providing the original image data: https://journals.plos.org/plosone/s/figures#loc-original-images-for-blots-and-gels.   In your cover letter, please note whether your blot/gel image data are in Supporting Information or posted at a public data repository, provide the repository URL if relevant, and provide specific details as to which raw blot/gel images, if any, are not available. Email us at plosone@plos.org if you have any questions. 5. Please review your reference list to ensure that it is complete and correct. If you have cited papers that have been retracted, please include the rationale for doing so in the manuscript text, or remove these references and replace them with relevant current references. Any changes to the reference list should be mentioned in the rebuttal letter that accompanies your revised manuscript. If you need to cite a retracted article, indicate the article’s retracted status in the References list and also include a citation and full reference for the retraction notice.

Reviewers' comments:

Reviewer's Responses to Questions

**Comments to the Author**

1. Is the manuscript technically sound, and do the data support the conclusions?

Reviewer #1: Partly

Reviewer #2: Partly

2. Has the statistical analysis been performed appropriately and rigorously? 

Reviewer #1: N/A

Reviewer #2: No

3. Have the authors made all data underlying the findings in their manuscript fully available?

Reviewer #1: Yes

Reviewer #2: Yes

4. Is the manuscript presented in an intelligible fashion and written in standard English?

Reviewer #1: Yes

Reviewer #2: Yes

5. Review Comments to the Author

Reviewer #1: The paper describes the design, expression, purification and testing of a fusion of the Pembrolizumab antibody with the IL15-IL15 complex.

Comments:

- Interesting idea, especially if individual co-administration studies have shown enhanced effect in tumour reduction.

- The novelty of the approach is in expressing the complex in plants and harvesting the same through farming.

Major Comments

- The functionality of the fused IL15alpha and IL15 is not described. In the absence of data on this part of the fusion, the entire hypothesis is questionable.

- ELISA on Pembrolizumab and the fusion binding- it was interesting to see that indeed the individual binding graphs has similiar slopes- a direct head to head competition assay would have thrown more light on the affinities

- The 'purified' molecule had multiple bands in the commassie blue staining but shows a single band on western. What are the bands then? Is this a purification artefact?

- The animal model- without details on the exposure data, PK (minimum) it is very difficult to conclude on the comparative efficacies- additionally without a dose response study a comparison of the two moieties would be incorrect.

- Wonder if the fusion protein is immunogenic- presence of antibody in the treated animals would have shed light on these aspects.

Reviewer #2: The manuscript entitled "Plant-derived Pembrolizumab in conjugation with IL-15Rα-IL-15 complex shows

improvement of anti-tumor activity" by Rattanapisit et al. is very interesting but I cannot fully agree with the conclusions based on the data.

First of all, the graphics don't show significant differences in binding or toxicity between the test molecule and keytruda.

Binding did not include error bars, thus it is not possible to decide whether the curves are significantly different. In fact, it is seen that the binding is a bit faster for keytruda, and maybe a little stronger for the test molecule, but without error bars and significance calculations it is not possible to make any conclusion. Yet, I would say that there is no difference in binding.

Second, tumor size and tumor volume don't show significant differences, while body weight seems a bit better for keytruda, although no real differences are seen at day 27. I think Figure 7 shows that the activity of both molecules is identical within the potential statistical deviations or experimental errors.

In conclusion, this is a good manuscript and the authors demonstrated that IL-15-conjugated Pembrolizumab can be successfully produced in a plant-based system, and as they have shown in their prior publications, the methods are sound and viable, and that conclusion is acceptable; but the conclusion that this conjugated molecule is better than the control molecule in controlling tumor growth is not supported by the data.

I would conclude that the manuscript can be accepted if the authors include all the statistical calculations for all the experiments, and either show evidence of their claim of better efficacy, or change their conclusion and the title, indicating that this construct is not significantly better than the test molecule; ie, keytruda.

6. PLOS authors have the option to publish the peer review history of their article (what does this mean?). If published, this will include your full peer review and any attached files.

Reviewer #1: **Yes: **Tanjore Soundararajan Balganesh

Reviewer #2: No

---

## [Author Response · Author response to Decision Letter 0]

16 Dec 2024

Responses to Reviewers’ comments

Reviewer #1: The paper describes the design, expression, purification and testing of a fusion of the Pembrolizumab antibody with the IL15-IL15 complex.

Comments:

- Interesting idea, especially if individual co-administration studies have shown enhanced effect in tumour reduction.

- The novelty of the approach is in expressing the complex in plants and harvesting the same through farming.

Response: Thank you for your positive feedback on this project. Within plant molecular farming platforms, we believe that this manuscript has demonstrated an important successful expression of antibody-cytokine fusion molecule using plants. The compound was produced steadily, and it showed acceptable anti-tumor activity, not different from the commercial Pembrolizumab.

Major Comments:

- The functionality of the fused IL15alpha and IL15 is not described. In the absence of data on this part of the fusion, the entire hypothesis is questionable.

Response: Thank you for your suggestion. We have included the functional description and importance of IL-15Rα and IL-15 parts in the introduction, line no. 42-45 and 64-66, and discussion, line no. 324-327. Basically, fusing IL-15Rα with IL-15 allows the compound to interact with IL-15Rβ and IL-15Rγ, located on the T and NK cells, without primarily binding to the antigen-presenting cells.

- ELISA on Pembrolizumab and the fusion binding- it was interesting to see that indeed the individual binding graphs has similar slopes- a direct head to head competition assay would have thrown more light on the affinities

Response: Thank you for your comment. We agreed that the head-to-head competition assay will assist direct comparison between the efficacy of unconjugated and IL-15-conjugated pembrolizumab. It will provide useful information on which compound would perform better in terms of PD-L1 protein binding. We have included this suggestion in the discussion, line no. 390-392, and aim to conduct this experiment in the next phase of product development. We have also revised the ELISA plot (Figure 4) to make it easier to understand.

- The 'purified' molecule had multiple bands in the Coomassie blue staining but shows a single band on western. What are the bands then? Is this a purification artefact?

Response: The bands that were observed at approximately 200-250 kDa on the SDS-PAGE gel, stained with Coomassie blue, but unfound on the Western blot analysis could be partial or cleaved pembrolizumab heavy chains. They might be co-purified with a complete form of pembrolizumab-IL-15Rα-IL-15 fusion by protein A affinity chromatography. After purification, these products appeared in the drug product and were detected with Coomassie blue staining due to protein being. However, they were not detected in the Western blot analysis with anti-Gamma and anti-Kappa antibodies because of incomplete formation of heavy chain and light chain antibody, lacking binding epitope for anti-Gamma and anti-Kappa antibodies to detect. We have incorporated this explanation in the discussion, line no. 361-367.

- The animal model- without details on the exposure data, PK (minimum) it is very difficult to conclude on the comparative efficacies- additionally without a dose response study a comparison of the two moieties would be incorrect.

Response: Thank you for your recommendation. We agreed that pharmacokinetics study and dose response curves would provide more insight into this study. We aim to conduct this pharmacokinetic study along with immunogenicity and extended efficacy in the next phase of product development. In this study, our primary goal is to demonstrate success of the design and production of Pembrolizumab fusion with IL-15Rα-IL-15 complex using the plant molecular farming system. We have added this suggestion to the discussion part, line no. 409-413.

- Wonder if the fusion protein is immunogenic- presence of antibody in the treated animals would have shed light on these aspects.

Response: Thank you for your valid suggestion. Yes, immunogenic data is crucial data to guarantee safety of this molecule. This part of the experiment will be carried out in the next phase of product development. In this study, we intended to show the design of the Pembrolizumab-IL-15Rα-IL-15 fusion and capability of plant molecular platform to produce such molecule. We have included this clarification in the discussion part, line no. 409-413.

Reviewer #2: The manuscript entitled "Plant-derived Pembrolizumab in conjugation with IL-15Rα-IL-15 complex shows improvement of anti-tumor activity" by Rattanapisit et al. is very interesting but I cannot fully agree with the conclusions based on the data.

Response: Thank you for your feedback. We have revised the conclusion of this manuscript per your suggestion. We, therefore, summarized that anti-tumor efficacy of unconjugated Pembrolizumab and Pembrolizumab-IL-15Rα-IL-15 was comparable in mouse colorectal cancer model. Nonetheless, the conjugated molecule showed a good sign of tumor-growth inhibition and should be further investigated for its potential and safety in different cancer models using dose response study.

First of all, the graphics don't show significant differences in binding or toxicity between the test molecule and keytruda. Binding did not include error bars, thus it is not possible to decide whether the curves are significantly different. In fact, it is seen that the binding is a bit faster for keytruda, and maybe a little stronger for the test molecule, but without error bars and significance calculations it is not possible to make any conclusion. Yet, I would say that there is no difference in binding.

Response: Thank you for pointing this out. Yes, we agreed that, with error bar and statistical data, the binding plot will improve data interpretation to discern the difference between the conjugated Pembrolizumab-IL-15Rα-IL-15 and Keytruda. We have edited Figure 4 by including error bars and statistical test for each timepoint. The tabular data was also supplied in Supplementary Table S2. We used two-way ANOVA with Sidak’s multiple comparison test to examine PD-1 binding efficacy between Pembrolizumab-IL-15Rα-IL-15 and Keytruda. Binding properties of both molecules were relatively similar. The only difference was observed at the concentration of 0.15 μg/ml, where Keytruda was slightly better bound to PD-1 protein. The calculated EC50 was not significant difference. Therefore, we concluded that the affinity to PD-1 protein of both molecules was relatively similar. We have included this point to the result, line no. 245-247.

Second, tumor size and tumor volume don't show significant differences, while body weight seems a bit better for keytruda, although no real differences are seen at day 27. I think Figure 7 shows that the activity of both molecules is identical within the potential statistical deviations or experimental errors.

Response: Thank you for your comment. We agreed that anti-tumor efficacy of unconjugated Pembrolizumab and Pembrolizumab-IL-15Rα-IL-15 were not significantly different. We toned down the efficacy results of Pembrolizumab-IL-15Rα-IL-15 and added suggestion for further studies to investigate its potential in different tumor models. In the manuscript, we have modified the abstract, line no 24-25, the result, line no. 294-295, the discussion, line no. 392-396 and 409-417 and the conclusion, line no. 422, to summarize that Pembrolizumab-IL-15Rα-IL-15 showed comparable tumor growth inhibition to commercial Keytruda.

In conclusion, this is a good manuscript and the authors demonstrated that IL-15-conjugated Pembrolizumab can be successfully produced in a plant-based system, and as they have shown in their prior publications, the methods are sound and viable, and that conclusion is acceptable; but the conclusion that this conjugated molecule is better than the control molecule in controlling tumor growth is not supported by the data.

I would conclude that the manuscript can be accepted if the authors include all the statistical calculations for all the experiments, and either show evidence of their claim of better efficacy, or change their conclusion and the title, indicating that this construct is not significantly better than the test molecule; ie, keytruda.

Response: Thank you for your appreciation of our study. We have revised the manuscript title to “Plant-derived Pembrolizumab in conjugation with IL-15Rα-IL-15 complex shows effective anti-tumor activity”. We have edited the discussion and conclusion to imply that this molecule showed acceptable anti-tumor efficacy and needs the following studies to understand its safety and efficacy profiles. In this manuscript, we highlight more on the design and production strategy of this molecule using plant molecular farming system. Moreover, we have included statistical data to Figure 4 as suggested. Statistical data for the other experiments was all provided.

---

## [Editor Report · Decision Letter 1]

17 Dec 2024

Plant-derived Pembrolizumab in conjugation with IL-15Rα-IL-15 complex shows effective anti-tumor activity

PONE-D-24-40114R1

Dear Dr. Suwanchaikasem,

We’re pleased to inform you that your manuscript has been judged scientifically suitable for publication and will be formally accepted for publication once it meets all outstanding technical requirements.

Kind regards,

Michael C Burger, M.D.

Academic Editor

PLOS ONE
---

## [Editor Report · Acceptance letter]

23 Dec 2024

PONE-D-24-40114R1 

PLOS ONE

Dear Dr. Suwanchaikasem, 

I'm pleased to inform you that your manuscript has been deemed suitable for publication in PLOS ONE. Congratulations! Your manuscript is now being handed over to our production team.

Kind regards, 

on behalf of

Dr. Michael C Burger 

Academic Editor

PLOS ONE